

# Phylogeny, time divergence, and historical biogeography of the South American *Liolaemus alticolor-bibronii* group (Iguania: Liolaemidae)

Sabrina N. Portelli[*] and Andrés S. Quinteros[*]

UNSa-CONICET, Instituto de Bio y Geociencias del NOA, Rosario de Lerma, Salta, Argentina
[*] These authors contributed equally to this work.

## ABSTRACT

The genus *Liolaemus* comprises more than 260 species and can be divided in two subgenera: *Eulaemus* and *Liolaemus sensu stricto*. In this paper, we present a phylogenetic analysis, divergence times, and ancestral distribution ranges of the *Liolaemus alticolor-bibronii* group (*Liolaemus sensu stricto* subgenus). We inferred a total evidence phylogeny combining molecular (*Cytb* and *12S* genes) and morphological characters using Maximum Parsimony and Bayesian Inference. Divergence times were calculated using Bayesian MCMC with an uncorrelated lognormal distributed relaxed clock, calibrated with a fossil record. Ancestral ranges were estimated using the Dispersal-Extinction-Cladogenesis (DEC-Lagrange). Effects of some *a priori* parameters of DEC were also tested. Distribution ranged from central Perú to southern Argentina, including areas at sea level up to the high Andes. The *L. alticolor-bibronii* group was recovered as monophyletic, formed by two clades: *L. walkeri* and *L. gracilis*, the latter can be split in two groups. Additionally, many species candidates were recognized. We estimate that the *L. alticolor-bibronii* group diversified 14.5 Myr ago, during the Middle Miocene. Our results suggest that the ancestor of the *Liolaemus alticolor-bibronii* group was distributed in a wide area including Patagonia and Puna highlands. The speciation pattern follows the South-North Diversification Hypothesis, following the Andean uplift.

## INTRODUCTION

The genus *Liolaemus* WIEGMAN 1834 currently includes 262 species (updated from (*Abdala, Quinteros & Semham, 2015*) distributed between Tierra del Fuego in southern Argentina and northern central Peru. The taxonomic composition and phylogenetic relationships of this genus have varied over the years. *Laurent (1983)* proposed dividing it into two main groups (subgenera): *Liolaemus sensu stricto* (or Chileno group) and *Eulaemus* (or Argentino group). Furthermore, taxonomic studies on these two groups have led to the recognition of numerous subgroups. Recently, *Troncoso-Palacios et al. (2015)* proposed the division of the *Liolaemus sensu stricto* subgenus into two main sections, the *L. chiliensis* section and the *L. nigromaculatus* section. The *L. alticolor-bibronii* group was attributed to the *Liolaemus sensu stricto* subgenus (*Espinoza, Wiens & Tracy,*

Corresponding author
Andrés S. Quinteros,
sebasquint@gmail.com

*2004*; *Lobo, 2005*; *Quinteros, 2012*; *Quinteros, 2013*) and the *L. chiliensis* section (*Troncoso-Palacios et al., 2015*; *Panzera et al., 2017*). The group was first defined by *Ortiz (1981)* as the *L. alticolor-walkeri* group, containing three species (*L. alticolor* BARBOUR 1909, *L. walkeri* SHREVE 1938, *and L. tacnae* SHREVE 1941), whereas the *L. bibronii* group was defined by *Cei (1986)* consisting of *L. bibronii* BELL 1843, *L. exploratorum* CEI & WILLIAMS 1984, and *L. sanjuanensis* CEI 1982. The taxonomic composition of these groups increased and varied in last years. Depending on the criteria, methods and dataset used, species were assigned to either more specific groups (*L. alticolor* or *L. bibronii* group— (*Lobo & Espinoza, 1999*, and *Martinez Oliver & Lobo, 2002*) or more inclusive groups, such as the *L. alticolor-bibronii* group (*Espinoza, Wiens & Tracy, 2004*; *Lobo, 2005*; *Quinteros, 2012*; *Aguilar et al., 2013*). In a contribution made by *Quinteros (2012)*, *L. alticolor* group was re-described including two new species resulting in a total number of 27 species. More recently, the number of species was corrected to 30 (*Martínez et al., 2011*; *Aguilar et al., 2013*; *Quinteros et al., 2014*; *Abdala, Quinteros & Semham, 2015*). *Morando et al. (2007)* performed a phylogeographic study on several populations of *L. bibronii* and *L. gracilis* BELL 1843, concluding that *L. bibronii* is in fact a species complex with many candidate species (some of them described by *Martínez et al. (2011)*, *Quinteros (2012)*, *Quinteros et al. (2014)* and *Abdala, Quinteros & Semham (2015)*). Using morphological characters *Quinteros (2013)* recovered the *L. alticolor-bibronii* group as monophyletic and found sister group relationships to *L. gravenhorsti*. Furthermore, *Aguilar et al. (2013)* described three new species from Peru and performed a phylogenetic analysis of local species of the *L. alticolor-bibronii* group. In summary, the *L. alticolor-bibronii* group is composed of 30 species, widely distributed in an outstanding altitudinal gradient from sea level to more than 5,000 masl (*Abdala & Quinteros, 2014*) between Santa Cruz Province (southern Argentina), Chile and Ancash (central Peru) (*Quinteros, 2012*; *Quinteros, 2013*; *Aguilar et al., 2013*). Due to these traits the *L. alticolor-bibronii* group is an excellent model to infer the historical events which possibly molded the current distribution of its members.

Until now, there have not been any proposals about the origin or diversification of the *L. alticolor-bibronii* group based on phylogenies or quantitative frameworks studying its biogeography. There are related studies, such as *Cei (1979)*, who characterized Patagonia as an active center of origin and dispersion, including *Liolaemus* as an example of a recent adaptive radiation. Other authors applied phylogenies, but failed to use explicit biogeographical methodology. *Lobo (2001)* showed a cladogram of the *chiliensis* group (*Liolaemus sensus stricto* subgenus) including distribution areas of the species. These areas are similar to those outlined by *Roig Juñent (1994)*, adding some, but not including others, such as the Puna. *Young-Downey (1998)* performed a Brooks Parsimony Analysis (BPA, *Brooks, 1990*) over a *Liolaemus* phylogeny, while *Schulte et al. (2000)* predicted species distribution based on molecular phylogeny. Even though no explicit methodology was included, the latter study can be considered an ancestral areas analysis. More details on the biogeography of the of the *Liolaemus sensu stricto* subgenus are given by *Díaz Gómez & Lobo (2006)* who applied three methods (Fitch optimization, DIVA—*Ronquist, 1997*, and Weighted Ancestral Areas Analysis—*Hausdorf, 1998*) to reconstruct the ancestral area of the subgenus.
In this study, we perform phylogenetic analyses in search of patterns which could have molded the current distribution of the species. We use quantitative and explicit methodology and estimate time divergence to investigate the historical biogeography of the *Liolaemus alticolor-bibronii* group.

## MATERIALS AND METHODS

### Taxon sampling and characters used

We included 30 species of the *Liolaemus alticolor-bibronii* group, taken primarily from *Quinteros (2013)* with recent species additions by *Martínez et al. (2011)*, *Aguilar et al. (2013)*, *Quinteros et al. (2014)* and *Abdala, Quinteros & Semham (2015)*, 15 populations of uncertain taxonomic status and eight outgroup species (increasing in nine the number of terminal taxa included by *Quinteros, 2013*). See File S1 for details.

The phylogenetic analyses included typical morphological characters (*Lobo, 2001*; *Lobo, 2005*; *Quinteros, 2012*; *Quinteros, 2013*) and sequences of C*ytb* and *12S* genes taken from the literature (*Espinoza, Wiens & Tracy, 2004*; *Morando et al., 2007*; *Victoriano et al., 2008*; *Aguilar et al., 2013*; *Troncoso-Palacios et al., 2016*; *Olave et al., 2011*; see File S1 for specimens' GenBank accession numbers). Sequences were aligned and edited with MEGA v.7.0.26 (*Kumar, Stecher & Tamura, 2016*). Morphological characters were coded into discrete and continuous based on their variation. Discrete characters were coded as Binary, Polymorphic Binary, Multistate, and Polymorphic Multistate, while continuous characters were coded "as such" following *Goloboff, Mattoni & Quinteros (2006)* methodology.

We included 960 bp from *12S*, 809 bp from C*ytb* and 167 morphological characters updating the list by *Quinteros (2013)*; see File S2 for details.

### Phylogenetic analyses

We conducted Maximum Parsimony (MP) and Bayesian Inference (BI) analyses using matrices including morphology + C*ytb* + *12S* characters. Alternative analyses excluding morphology and/or continuous characters were also performed.

Maximum Parsimony analyses were implemented in TNT v.1.1 (*Goloboff, Farris & Nixon, 2003*; *Goloboff, Farris & Nixon, 2008*; *Goloboff & Catalano, 2016*) under equal and implied weights (Concavity value = 3; 4; 5; and 6). Continuous characters were included in MP analyses only, since TNT is the only software supporting them. Runs were performed using traditional search Tree Bisection Reconnection (TBR), with 500 replicates and saving 20 trees each. Additionally, we performed an analysis using the New Technology Search tool implemented in TNT (Sectorial Search, Tree Fusing, Tree Drifting, and Ratchet) with 50 replicates, hitting the best score at least 20 times. Support was measured under Symmetric Resampling with 500 replicates and a 33% deletion.

For BI, we selected the best-fitting model using jModel Test 3.0.4 (*Posada, 2008*). The best-fitting model for CytB and 12S individually was GTR+G, whereas GTR (GTR+Γ+I) fitted best for the concatenate genes. Bayesian Inference analyses were carried out in Mr. Bayes v3.1 (*Ronquist & Huelsenbeck, 2003*). We run PartitionFinder v1.1.1 (*Lanfear et al., 2012*) to detect the best partition scheme, including both genes (without partition matrix). Calculations were run twice for 10 million generations each, resulting in a final average

standard deviation of Split frequencies below 0.05, sampling trees every 1,000 generations and using four simultaneous chains (one cold and three hot) in each run. Convergences of the chain to stationary distribution were confirmed using Tracer v1.6 (*Rambaut et al., 2014*). We discarded as burn-in the first one thousand sampled trees that were not within the stationary distribution of log likelihoods. Trees and posterior probabilities were summarized using the "50% majority rule" consensus method.

## Time divergence estimates

Age of nodes and substitution rates were simultaneously estimated (for both topologies, BI and MP) using Bayesian MCMC (Marcov Chain Monte Carlo) approach as implemented in BEAST v2.4.0 (*Drummond & Rambaut, 2007*). We used a fossil assigned to *Eulaemus* (*Albino, 2008*), representing the earliest record of this subgenus, to place a mean prior of 20 Myr on the tree height. Since the fossil was assigned to the *Eulaemus* subgenus, without specific status, in our study we must place it as an outgroup taxon, sister of the species members of the *Liolaemus* sensu stricto subgenus, which are the focal species. Divergence times in BEAST were estimated according to *Fontanella et al. (2012)*: (1) a lognormal prior was employed for fossil calibrations (*Hedges & Kumar, 2004*); (2) a Yule speciation process with a random starting tree was used for the tree prior; (3) an uncorrelated lognormal distributed relaxed clock (UCLD) model was employed, allowing evolutionary rates to vary along branches within lognormal distributions (*Drummond et al., 2006*). Three independent runs of 50.000.000 generations each were performed, with sampling every 5,000 generations. The three separate runs were then combined (following removal of 10% burn-in) using Log Combiner v2.0 (*Drummond & Rambaut, 2007*). Adequate sampling and convergence of the chain to stationary distribution were confirmed by inspection of MCMC samples using Tracer v1.6 (*Rambaut et al., 2014*). The effective sample size (ESS) values of all parameters were greater than 200, which was considered a sufficient level of sampling. The sampled posterior trees were summarized using Tree Annotator v2.0 (*Drummond & Rambaut, 2007*) to generate a maximum clade credibility tree (maximum posterior probabilities) and to calculate the mean ages, 95% highest posterior density (HPD) intervals, posterior probabilities and substitution rates for each node. The BEAST topology was visualized with Fig Tree v1.2 (*Rambaut, 2008*).

## Biogeographical analyses

In order to assign the ancestral geographic areas of distribution, the regionalization of South America according to *Morrone (2001)* was used. Briefly, *Morrone (2001)* detailed biogeographic regions, sub regions, and provinces of Latin America and the Caribbean, pointing out characteric taxa and their predominant vegetation. Even though those areas are defined by contemporary species, they are a useful methodological tool to avoid the use of randomly defined areas. We chose the following provinces based on the georeferenced distribution of the species included (Fig. 1): A: Desierto Peruano Costero, B: Puna, C: Yungas, D: Atacama, E: Coquimbo, F: Prepuna, G: Monte, H: Chaco, I: Pampa, J: Santiago, K: Maule and L: Patagonia Central. To infer the processes which modeled the current distribution of the species of the *L. alticolor-bibronii* group we used a parametric method
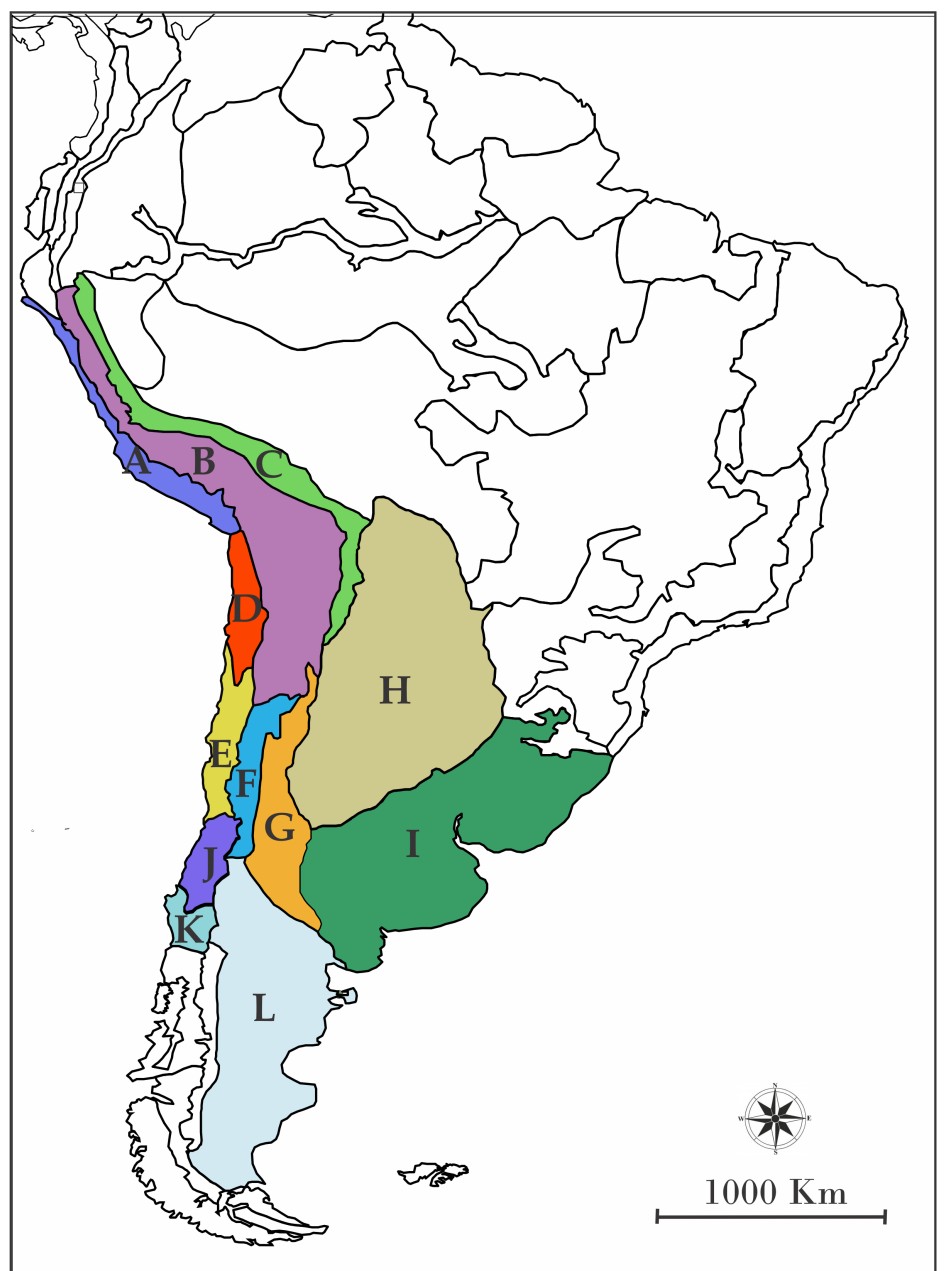

**Figure 1  Map of South America showing the biogeographic regions used.** Biogeographic regions of *Morrone (2001)* employed in DEC analyses. (A) Desierto Peruano Costero, (B) Puna, (C) Yungas, (D) Atacama, (E) Coquimbo, (F) Prepuna, (G) Monte, (H) Chaco, (I) Pampa, (J) Santiago, (K) Maule, (L) Patagonia Central, and (M) Patagonia Subandina.

implemented in RASP (*Nylander et al., 2008*; *Yu, Harris & He, 2010*). This method is based on Dispersal-Extinction-Cladogenesis (DEC) models which require information about a single ultrametric-dated phylogeny and distributional information of extant species.

For DEC we used the two calibrated trees recovered in BEAST (based on BI and MP topologies) and applied two different adjacency matrices: (1) unconstrained and (2)

**Table 1  Probabilities of dispersal cost.** Probabilities between the areas (A–L) employed in the DEC analysis under Matrix II. See 'Materials and Methods' for area names.

|   | A | B | C | D | E | F | G | H | I | J | K | L |
|---|---|---|---|---|---|---|---|---|---|---|---|---|
| A | 1 | 0.8 | 0.6 | 0.8 | 0.6 | 0.4 | 0.6 | 0.6 | 0.4 | 0.4 | 0.2 | 0.4 |
| B | 0.8 | 1 | 0.8 | 0.8 | 0.8 | 0.8 | 0.8 | 0.8 | 0.6 | 0.6 | 0.4 | 0.6 |
| C | 0.6 | 0.8 | 1 | 0.6 | 0.6 | 0.6 | 0.6 | 0.8 | 0.6 | 0.4 | 0.2 | 0.4 |
| D | 0.8 | 0.8 | 0.6 | 1 | 0.8 | 0.6 | 0.6 | 0.6 | 0.4 | 0.6 | 0.4 | 0.4 |
| E | 0.6 | 0.8 | 0.6 | 0.8 | 1 | 0.8 | 0.6 | 0.4 | 0.4 | 0.6 | 0.6 | 0.6 |
| F | 0.4 | 0.8 | 0.6 | 0.6 | 0.8 | 1 | 0.8 | 0.6 | 0.6 | 0.8 | 0.6 | 0.8 |
| G | 0.4 | 0.8 | 0.6 | 0.6 | 0.6 | 0.8 | 1 | 0.8 | 0.8 | 0.6 | 0.6 | 0.8 |
| H | 0.2 | 0.8 | 0.8 | 0.6 | 0.4 | 0.6 | 0.8 | 1 | 0.8 | 0.4 | 0.4 | 0.6 |
| I | 0.2 | 0.6 | 0.6 | 0.4 | 0.4 | 0.6 | 0.8 | 0.8 | 1 | 0.4 | 0.6 | 0.8 |
| J | 0.4 | 0.6 | 0.4 | 0.6 | 0.6 | 0.8 | 0.6 | 0.4 | 0.4 | 1 | 0.8 | 0.8 |
| K | 0.2 | 0.4 | 0.2 | 0.4 | 0.6 | 0.6 | 0.6 | 0.4 | 0.6 | 0.8 | 1 | 0.8 |
| L | 0.4 | 0.6 | 0.4 | 0.4 | 0.6 | 0.8 | 0.8 | 0.6 | 0.8 | 0.8 | 0.8 | 1 |

**Table 2  Probabilities of dispersal cost.** Probabilities between the areas (A–L) employed in the DEC analysis under Matrix III. See 'Materials and Methods' for area names.

|   | A | B | C | D | E | F | G | H | I | J | K | L |
|---|---|---|---|---|---|---|---|---|---|---|---|---|
| A | 1 | 0.001 | 0.001 | 0.8 | 0.6 | 0.001 | 0.001 | 0.001 | 0.001 | 0.4 | 0.2 | 0.001 |
| B | 0.001 | 1 | 0.8 | 0.001 | 0.001 | 0.8 | 0.8 | 0.8 | 0.6 | 0.001 | 0.001 | 0.6 |
| C | 0.001 | 0.8 | 1 | 0.001 | 0.001 | 0.6 | 0.6 | 0.8 | 0.6 | 0.001 | 0.001 | 0.4 |
| D | 0.8 | 0.001 | 0.001 | 1 | 0.8 | 0.001 | 0.001 | 0.001 | 0.001 | 0.6 | 0.4 | 0.001 |
| E | 0.6 | 0.001 | 0.001 | 0.8 | 1 | 0.001 | 0.001 | 0.001 | 0.001 | 0.8 | 0.6 | 0.001 |
| F | 0.001 | 0.8 | 0.6 | 0.001 | 0.001 | 1 | 0.8 | 0.6 | 0.6 | 0.001 | 0.001 | 0.8 |
| G | 0.001 | 0.8 | 0.6 | 0.001 | 0.001 | 0.8 | 1 | 0.8 | 0.8 | 0.001 | 0.001 | 0.8 |
| H | 0.001 | 0.8 | 0.8 | 0.001 | 0.001 | 0.6 | 0.8 | 1 | 0.8 | 0.001 | 0.001 | 0.6 |
| I | 0.001 | 0.6 | 0.6 | 0.001 | 0.001 | 0.6 | 0.8 | 0.08 | 1 | 0.001 | 0.001 | 0.8 |
| J | 0.4 | 0.001 | 0.001 | 0.6 | 0.8 | 0.001 | 0.001 | 0.001 | 0.001 | 1 | 0.8 | 0.001 |
| K | 0.2 | 0.001 | 0.001 | 0.4 | 0.6 | 0.001 | 0.001 | 0.001 | 0.001 | 0.8 | 1 | 0.001 |
| L | 0.001 | 0.6 | 0.6 | 0.001 | 0.001 | 0.8 | 0.8 | 0.6 | 0.8 | 0.001 | 0.001 | 1 |

constrained (See 'Discussion' for details). We also included three different "area-dispersal" matrices: a first analysis (Matrix I) was set to allow a dispersal event between every geographic area without any cost; a second analysis (Matrix II) was set to apply dispersal cost, where we randomly assigned probability values of dispersion between areas, taking into account the closeness between them. The values were: 0.8 for adjacent areas; 0.6 for areas separated by one intermediate area; 0.4 for areas separated by two intermediate areas; 0.2 for areas separated by three intermediate areas; and 0.08 for areas separated by four intermediate areas (Table 1). A third analysis (Matrix III) was set to apply dispersal cost plus geographic barriers cost, considering probability values from the second analysis plus values assigned to geographic barriers (in this case, the only clear geographical barrier was the Andes mountain range). When a geographic barrier was present, a probability value of close to 0 was assigned (0.001—Table 2).
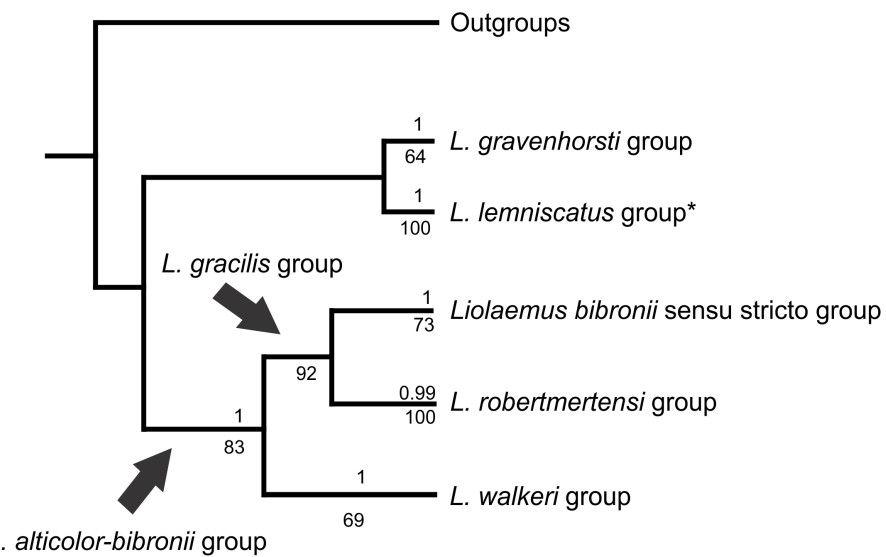

**Figure 2** **Congruence between the Maximum Parsimony (MP) and Bayesian Inference (BI) analyses** *
Some members of the *L. lemniscatus* clade is recovered as a sister group of the *L. walkeri* clade under MP.
Numbers above nodes correspond to Posterior Probabilities (Only Probabilities over 0.95 are shown).
Numbers under nodes correspond to Symmetric Resampling values.

## RESULTS

### Phylogeny of the *Liolaemus alticolor-bibronii* group

The analyses showed two topologies: one recovered under MP and the other under BI.
The topologies found are congruent in the main groups (SPR distance 0.6277—Fig. 2);
however, we also detected incongruence within them.

The *Liolaemus alticolor-bibronii* group was recovered in both MP and BI analyses, with
different composition and with high support (Figs. 3 and 4). In the BI tree, a clade formed
by the *L. gravenhorsti* group and the *L. lemniscatus* group was recovered as sister of the
*L. alticolor-bibronii* group. On the other hand, the MP analysis recovered the *L. lemniscatus*
group as a member of the *L. alticolor-group*, but some terminal taxa were not recovered as
members, despite their previously assignment to the *L. alticolor-bibronii* group (*L. paulinae*
DONOSO BARROS 1961, *L.* sp4, *L.* sp5, and *L.* sp14).

Within the *Liolaemus alticolor-bibronii* group, two main groups were recovered: the
*L. gracilis* and *L. walkeri* group (Figs. 2–4). The *L. walkeri* group, formed by terminal taxa
of northern distribution ranges (North Argentina, Bolivia and Peru), was recovered with
high support in BI, but moderate MP support. The *L. gracilis* group, on the other hand, was
recovered with high support in MP, but low support in BI. The latter group contains two
clades: (1) a clade formed by species distributed in central-southern Argentina (referred to
as *L. bibronii sensu stricto* clade) and (2) the *L. robertmertensi* group.

The taxonomic composition of each group recovered under MP and BI is listed in
Table 3.

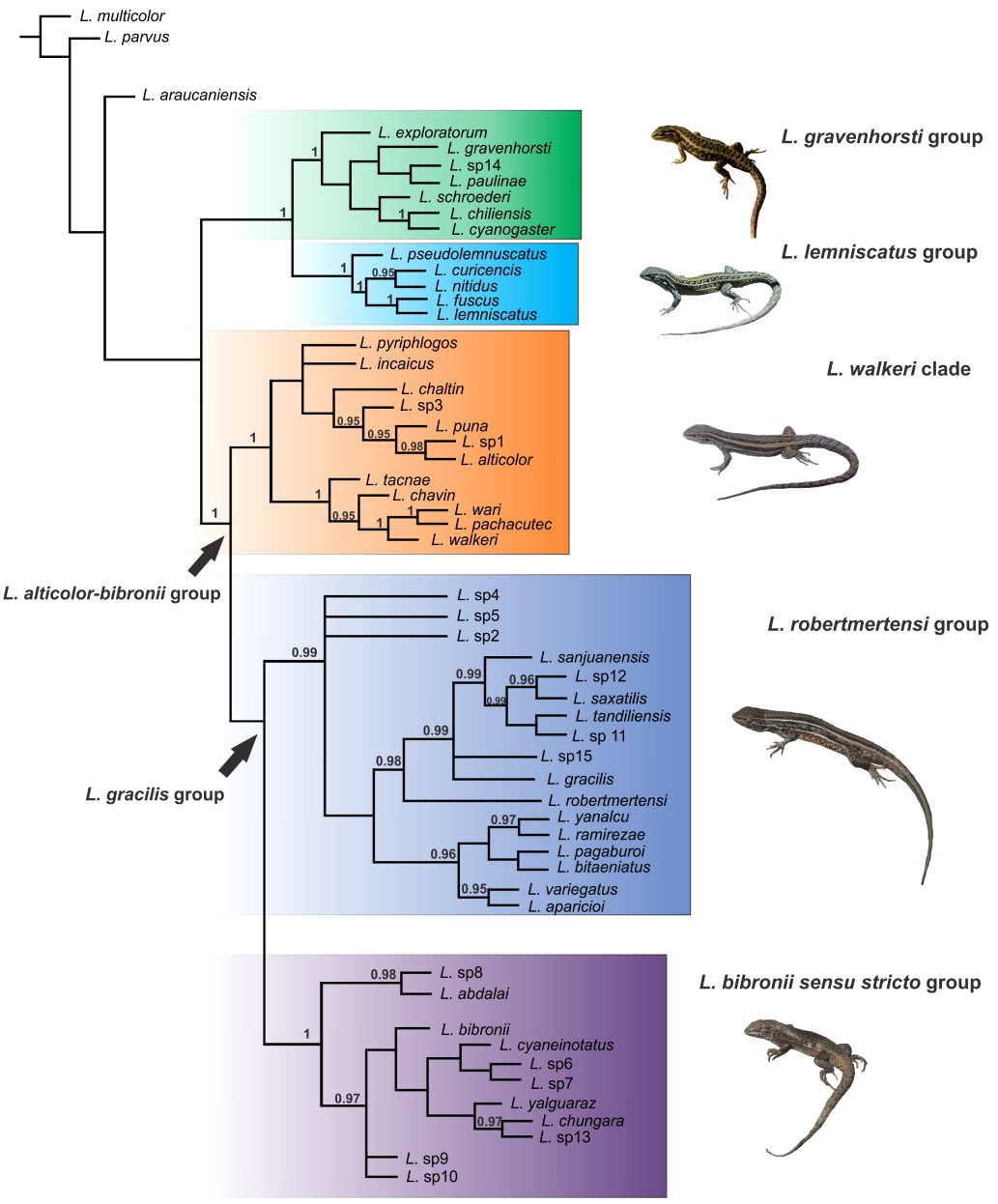

**Figure 3** **Topology recovered with Bayesian Inference.** Numbers above nodes correspond to Posterior Probability values (only values over 0.95 are shown).

As we mentioned above, *Liolaemus paulinae,* previously considered member of the *L. alticolor-bibronii* group, was not recovered as a member of the group. The same topology was recovered for *L.* sp14 (sister taxon of *L. paulinae*). *Liolaemus araucaniensis* MÜLLER & HELLMICH 1932 did not show relations to any group, suggesting this species could be related to the *L. belli* group.

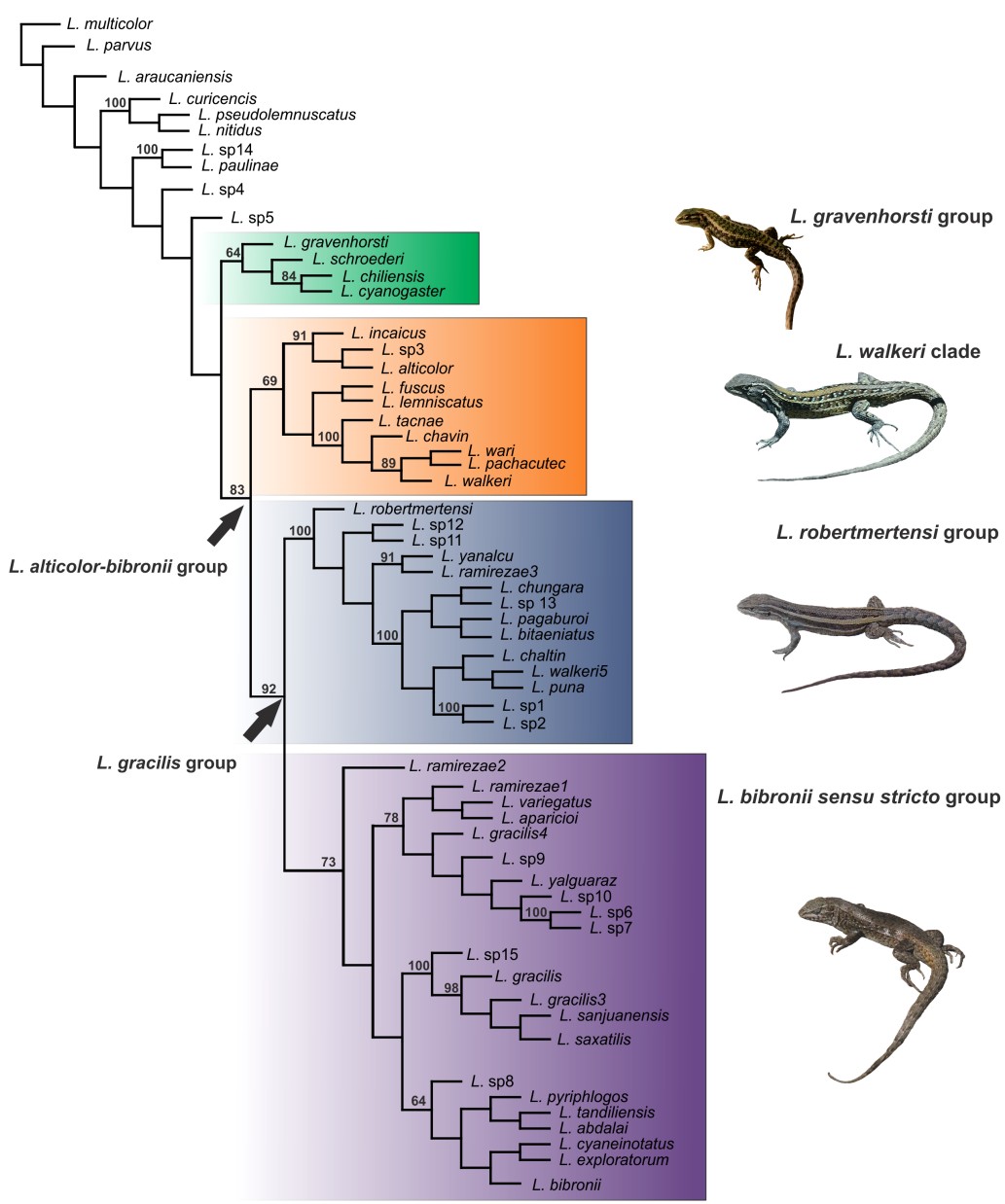

**Figure 4  Topology recovered with Maximum Parsimony.** Numbers above nodes correspond to Symmetric Resampling values (only values over 60 are shown). Note that the *L. lemniscatus* is paraphyletic and some species of this group are nested inside the *L. walkeri* clade.

### Divergence time estimates

Divergence time estimates of the *Liolaemus alticolor-bibronii* group were obtained using a time-calibrated tree with BEAST including MP and BI topologies (Figs. 5 and 6).

### Dating

The two topologies resulted in two time calibrated trees with similar times of divergence. Our results suggest that, according to the BI topology, the *Liolaemus gravenhorsti* group

**Table 3  Taxonomic composition of the main groups.** Species and populations members of the main groups recovered inside the *Liolaemus alticolor-bibronii* group under Maximum Parsimony and Bayesian Inference.

| Group | Maximum Parsimony | Bayesian Inference |
|---|---|---|
| *L. gravenhorsti* | *L. chiliensis; L. cyanogaster; L. gravenhorsti; L. schroederi* | *L. chiliensis; L. cyanogaster; L. exploratorum; L. gravenhorsti; L. paulinae; L. schroederi* |
| *L. lemniscatus* | *L. fuscus; L. lemniscatus* | *L. curicencis; L. fuscus; L. lemniscatus; L. nitidus; L. pseudolemniscatus* |
| *L. alticolor-bibronii* | *L. gracilis* group *L. walkeri* group | *L. gracilis* group *L. walkeri* group |
| *L. walkeri* | *L. alticolor; L. chavin; L. incaicus; L. lemniscatus* group; *L. pachacutec; L.* sp3; *L. tacnae; L. walkeri; L. wari* | *L. alticolor; L. chaltin; L. chavin; L. incaicus; L. pachacutec; L. puna; L. pyriphlogos* |
| *L. gracilis* | *L. bibronii sensu stricto* group *L. robertmertensi* group | *L. bibronii sensu stricto* group *L. robertmertensi* group *L.* sp2; *L.* sp4; *L.* sp5 |
| *L. bibronii* sensu stricto | *L. abdalai; L. aparicioi; L. bibronii; L. cyaneinotatus; L. exploratorum; L. gracilis; L. gracilis*3; *L. gracilis*4; *L. pyriphlogos; L. ramirezae* 1; *L. ramirezae*2; *L. sanjuanensis; L. saxatilis; L.* sp6; *L.* sp7; *L.* sp8; *L.* sp9; *L.* sp10; *L.* sp15; *L. tandiliensis; L. variegatus; L. yalguaraz* | *L. abdalai; L. bibronii; L. chungara; L. cyaneinotatus; L.* sp6; *L.* sp7; *L.* sp8; *L.* sp9; *L.* sp10; *L.* sp13; *L. yalguaraz* |
| *L. robertmertensi* | *L. bitaeniatus; L. chaltin; L. chungara; L. pagaburoi; L. puna; L. ramirezae* 3; *L. robertmertensi; L.* sp1; *L.* sp2; *L.* sp11; *L.* sp12; *L.* sp13; *L. walkeri* 5; *L. yanalcu* | *L. aparicioi; L. bitaeniatus; L. gracilis; L. pagaburoi; L. ramirezae; L. robertmertensi; L. sanjuanensis; L. saxatilis; L.* sp11; *L.* sp12; *L.* sp15; *L. tandiliensis; L. variegatus; L. yanalcu* |

(sister group of the *L. alticolor-bibronii* group) diverged in the Early Miocene 7 Myr (95% highest posterior density interval (HPD): 9.32–4.89) ago, whereas in the MP topology this clade diverged 3.25 Myr (95% HPD: 4.82–1.71) ago. This clade is formed by species distributed in Argentina and Chile.

We estimated the divergence time of the *Liolaemus alticolor-bibronii* group to Middle Miocene around 12.84 Myr ago for BI (95% HPD: 20.01–5.64) and 14.05 Myr ago for MP (95% HPD: 23.56–4.54).

Considering the clades of the *Liolaemus alticolor-bibronii* group, we estimated that the *L. gracilis* clade diverged 10.49 Myr (95% HPD: 17.06–3.85) ago for BI, being 12.39 Myr (95% HPD: 17.42–7.25) for MP. This clade splits into two clades: the *L. bibronii sensu stricto* clade and the *L. robertmertensi group*. The *L. bibronii sensu stricto* clade initiated its divergence 4.68 Myr (95% HPD: 7.25–2.11) ago in BI, whereas for MP this clade diverged 9.82 Myr (95% HPD: 11.85–5.78) ago. The *L. robertmertensi* group does so 9.37 Myr (95% HPD: 15.68–3.01) ago in BI, and 4.2 Myr (95% HPD: 6.84–1.67) ago for MP topology. The latter originated two clades, one distributed in central-southern Argentina, which diverged 6.75 Myr (95% HPD: 10.29–3.23) ago and the other distributed in Northwestern Argentina and Bolivia, diverging 5.62 Myr (95% HPD: 7.45–3.81) ago, these clades are recovered only under BI.

The *Liolaemus walkeri* clade dates back to the Late Miocene, around 11.54 Myr (95% HPD: 20.26–2.8) ago in our BI topology, 12.78 Myr (95% HPD: 19.65–5.87) ago in MP. *Liolaemus walkeri* group includes a clade distributed exclusively in Peru (with an origin estimated at 10.38 Myr—95% HPD: 18.15–2.55 in BI, and 10.72 Myr—95% HPD: 19.32–2.12 for MP) as well as a clade distributed mainly in Bolivia (origin estimated around 2.45 Myr—95% HPD: 4.26–0.62—recovered only with BI). While the topology recovered

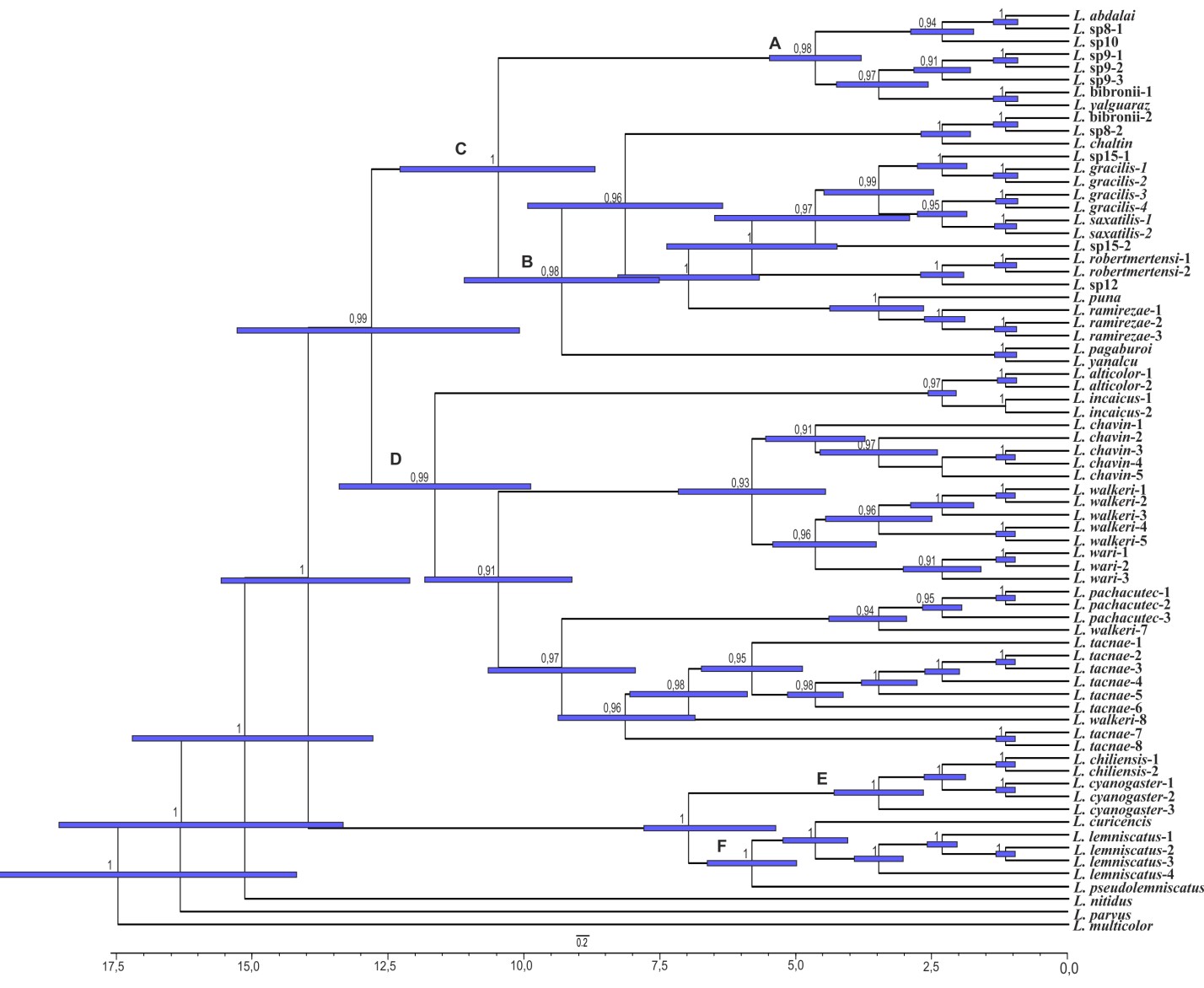

**Figure 5** **Times of divergences estimates for the *L. alticolor-bibronii* group, under BI topology** Ultrametric tree scaled in Myr. Numbers and horizontal bars on nodes represent posterior probabilities values and 95% credibility intervals. (A) *L. bibronii sensu stricto* group; (B) *L. robertmertensi* group; (C) *L. gracilis* group; (D) *L. walkeri* clade; C + D: *L. alticolor-bibronii* group (E) *L. gravenhorsti* group; (F) *L. lemniscatus* group.



under MP *L. lemniscatus* diverged 3.15 Myr (95% HPD: 5.05–1.23) ago from the *L. walkeri* clade, the BI topology, shows the *L. lemniscatus* group as a member of the *L. gravenhorsti* group (outside of the *L. alticolor-bibronii* group) diverging 3.5 Myr (95% HPD: 5.85–1.22) ago.

## Reconstruction of ancestral distribution

We compared six different scenarios in DEC, for each topology. The best Likelihood scores were obtained with the unconstrained adjacency matrix (Table 4). Based on this, we show the results recovered with the unconstrained adjacency matrix for both topologies (BI and

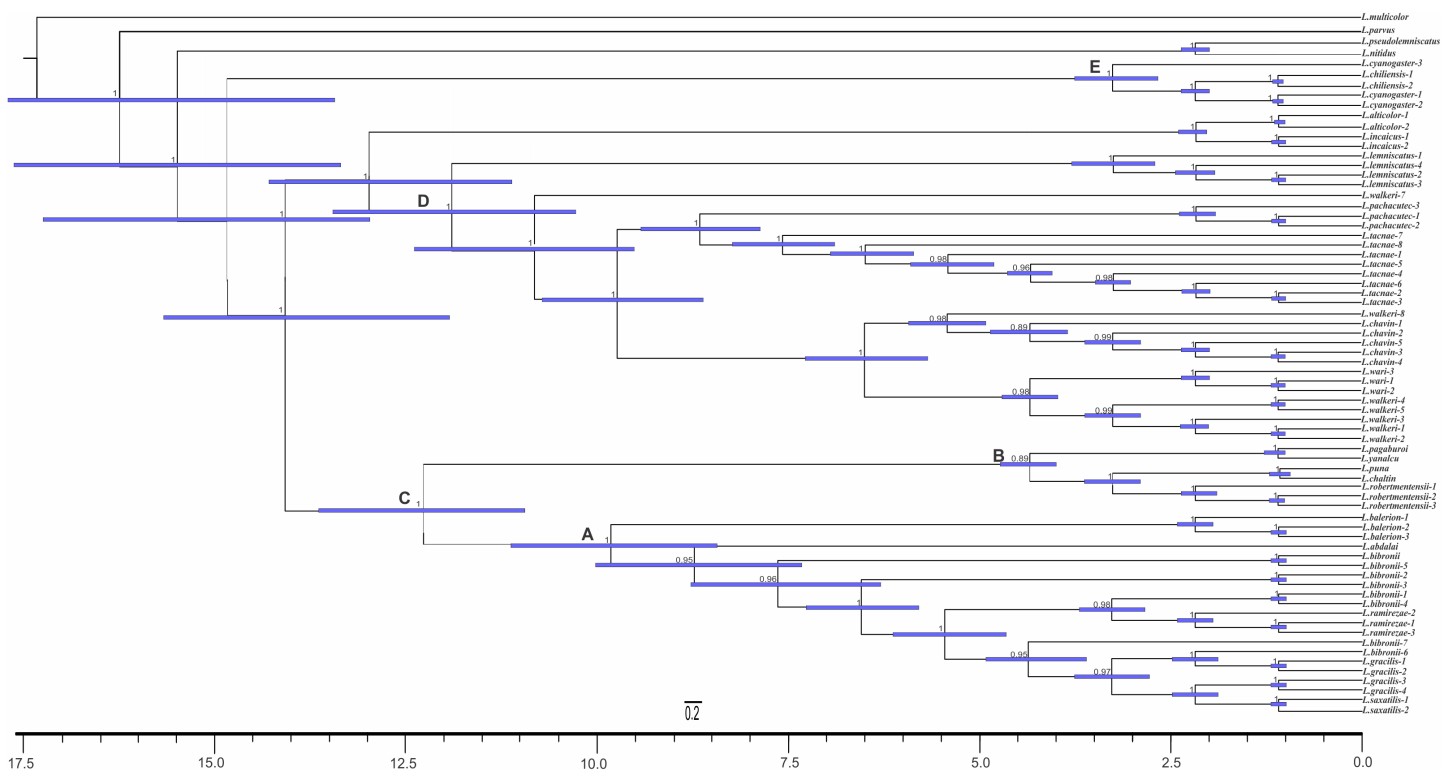

**Figure 6** **Times of divergences estimates for the *L. alticolor-bibronii* group, under MP topology.** Ultrametric tree scaled in Myr. Numbers and horizontal bars on nodes represent posterior probabilities values and 95% credibility intervals. (A) *L. bibronii sensu stricto* group; (B) *L. robertmertensi* group; (C) *L. gracilis group*; (D) *L. walkeri* clade; C + D: *L. alticolor-bibronii* group (E) *L. gravenhorsti* group.

MP). We obtained the same biogeographic scenarios for all three dispersion cost matrices used (Figs. 7 and 8). Alternative scenarios obtained with the constrained adjacency matrix can be found in File S3.

Divergence times, ancestral range and major probabilities of the main groups recovered are shown in Table 5.

The ancestral area of the *Liolaemus alticolor-bibronii* and the *L. gravenhorsti* groups correspond to the range BJ (Puna and Santiago; p: (1) and FJ (Prepuna and Santiago; p: 0.65) for the BI and MP topology, respectively. Due to dispersion and/or vicariance events, the *L. alticolor-bibronii* group occupies areas in Argentina, whereas the *L. gravenhorsti* remains confined to Chile.

Our results suggest that the ancestral area of the *Liolaemus alticolor-bibronii* group is formed by the areas BL (Puna and Patagonia Central; p: 0.67–BI) and F (Prepuna; p: 0.41–MP). Subsequently, a vicariance event for BI, and a dispersal event (MP) originated the *L. gracilis* group and the *L. walkeri* clade. With lower probabilities the analyses shows the following ancestral areas for the *L. alticolor-bibronii* group: B (Puna; p: 0.16) for BI; and BF (Puna and Prepuna; p: 0.26) for MP; the range BH (Puna and Chaco; p: 0.17) for BI, and FJ (Prepuna and Santiago; p: 0.23) or FL (Prepuna and Patagonia Central; p: 0.09) for MP.
**Table 4  Comparison between the results obtained in DEC applying two different adjacency matrices.** Number of alternative ancestral areas obtained from the main groups applying constrained and unconstrained adjacency matrix. MI, MII, and MIII correspond to dispersion coast and geographic barriers matrix (see Materials and Methods for details). Global maximum Likelihood scores (−lnL) and dispersal and extinction rates estimated.

|  |  | Global dispersal | Global vicariance | Global extinction |
|---|---|---|---|---|
| **Bayesian Inference** |  |  |  |  |
| Constrained adjancency matrix | Matrix I | 33 | 10 | 0 |
|  | Matrix II | 39 | 23 | 8 |
|  | Matrix III | 46 | 22 | 12 |
| Unconstrained adjancency matrix | Matrix I | 31 | 16 | 0 |
|  | Matrix II | 36 | 16 | 2 |
|  | Matrix III | 31 | 16 | 0 |
| **Parsimony** |  |  |  |  |
| Constrained adjancency matrix | Matrix I | 37 | 18 | 1 |
|  | Matrix II | 35 | 21 | 6 |
|  | Matrix III | 37 | 22 | 6 |
| Unconstrained adjancency matrix | Matrix I | 37 | 16 | 0 |
|  | Matrix II | 36 | 21 | 6 |
|  | Matrix III | 37 | 20 | 6 |

|  |  | −lnL | Dispersal rates | Extinction rates |
|---|---|---|---|---|
| **Bayesian Inference** |  |  |  |  |
| Constrained adjancency matrix | Matrix I | 156.41 | 0.024 | 0.013 |
|  | Matrix II | 154.22 | 0.015 | 0.018 |
|  | Matrix III | 161.32 | 0.022 | 0.020 |
| Unconstrained adjancency matrix | **Matrix I** | **147.91** | **0.012** | **0.002** |
|  | **Matrix II** | **147.47** | **0.009** | **0.008** |
|  | **Matrix III** | **145.54** | **0.015** | **0.021** |
| **Parsimony** |  |  |  |  |
| Constrained adjancency matrix | Matrix I | 196.25 | 0.011 | 0.013 |
|  | Matrix II | 167.96 | 0.021 | 0.026 |
|  | Matrix III | 176.70 | 0.035 | 0.025 |
| Unconstrained adjancency matrix | **Matrix I** | **165.75** | **0.004** | **0.012** |
|  | **Matrix II** | **163.39** | **0.008** | **0.014** |
|  | **Matrix III** | **157.28** | **0.002** | **0.021** |

The *Liolaemus gracilis* group has its origins in L (Patagonia Central; p: 0.5) for both topologies. A dispersion event inside the *L. gracilis* group, originated the *L. robertmertensi* and the *L. bibronii* groups. The first one, shows the HL range (Chaco and Patagonia Central; p: (1) as its ancestral area for BI and area B (Puna; p: 0.36) for MP, whereas the *L. bibronii sensu stricto* remains in L: Patagonia Central ($p = 1$ BI; $p = 0.57$ MP).

The *Liolaemus walkeri* group has B (Puna; p: 1) as its ancestral area for BI, and BF (Puna and Prepuna; p: 0.87) for MP. After a dispersion event (BI) and a vicariance event (MP), the *L. walkeri* group split into two clades. One occupied the range BC (Puna and Yungas, p: 0.84) in BI, and B (Puna; p: 1) in MP, and the other remained in B (Puna; p: 0.68) for
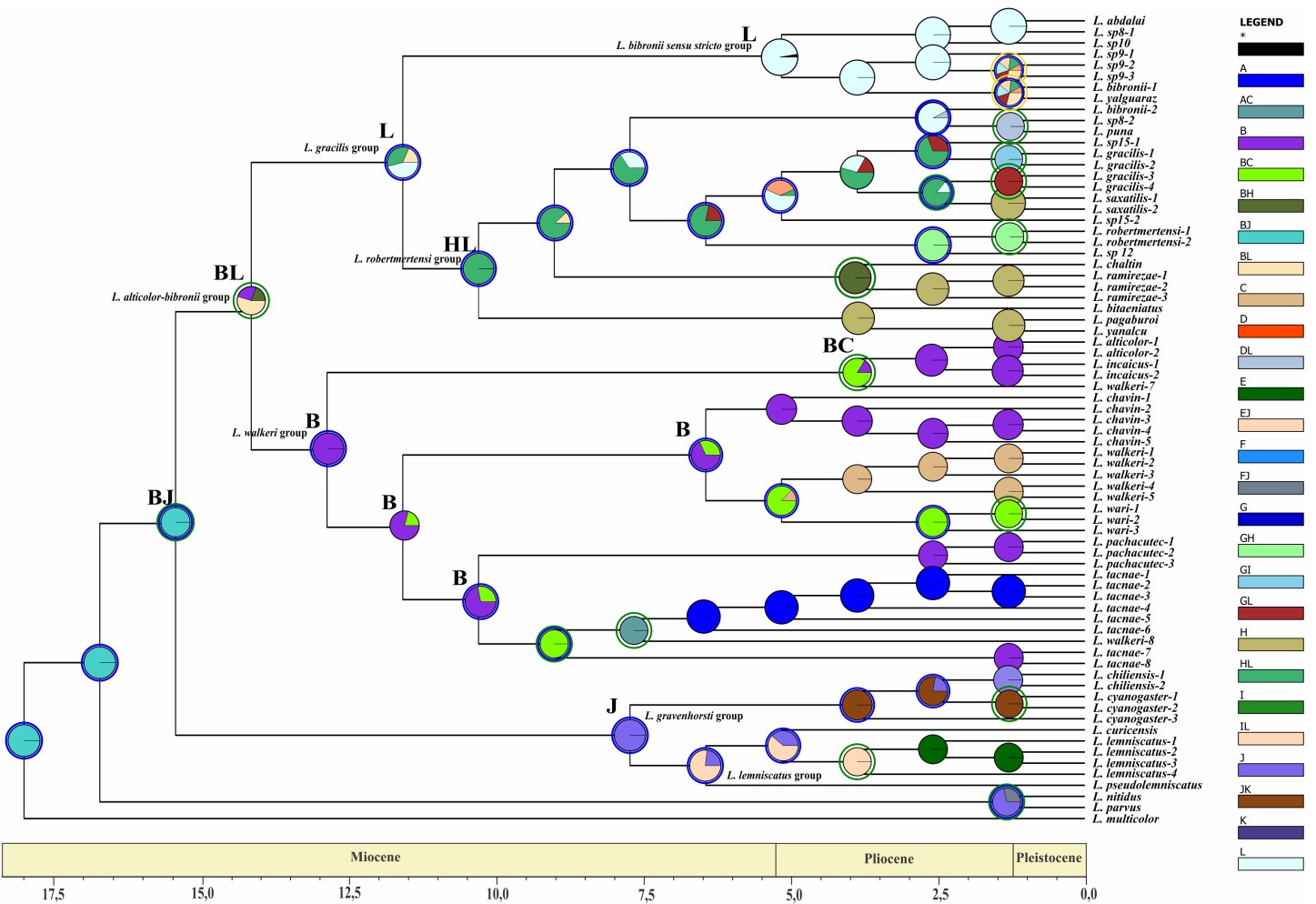

**Figure 7** **Ancestral area of distribution (unconstrained adjacency matrix-BI topology).** The analysis was run by randomly dividing the entire time span of evolution of the *Liolaemus alticolor- bibronii* group in three periods (18–5.5 Myr; 5.5–1.25 Myr; 1.25 Myr-present). Pie charts of each node depict the relative probabilities of ancestral area/ranges (showed in Legend). Letters above nodes represent ancestral ranges. Circles around pie charts represent events: blue circle: dispersal event; green circle: vicariance. See material and methods for area names shown in South America map. Time axis (in Myr) is annotated with major geological events.

both topologies. In the topology recovered under MP, the *L. walkeri* clade includes the *L. lemniscatus* group, whose ancestral area corresponds to the EF range (Coquimbo and Prepuna; p: 1), but after an extinction event is currently restricted to E (Coquimbo).

# DISCUSSION

## Phylogeny of the *Liolaemus alticolor-bibronii* group

We recovered the *Liolaemus gravenhorsti* group as sister taxon of the *Liolaemus alticolor-bibronii* group. These results are in accordance with previous morphological— (*Lobo, 2005*; *Díaz Gómez & Lobo, 2006*; *Quinteros, 2013*) and molecular—(*Schulte et al., 2000*; *Schulte, 2013*; *Pyron, Burbrink & Wiens, 2013*; *Zheng & Wiens, 2016*) based phylogenies.
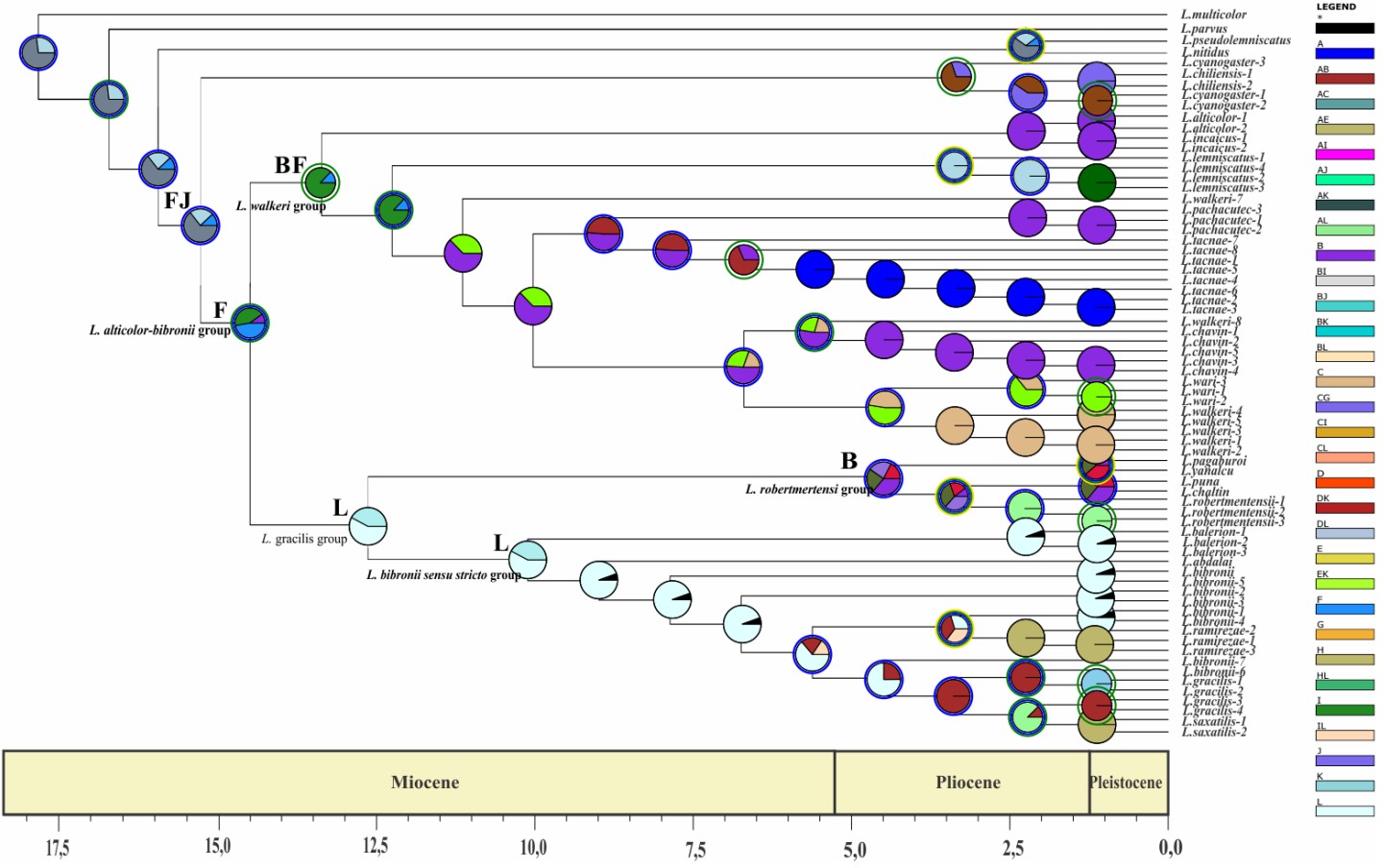

**Figure 8** **Ancestral area of distribution (unconstrained adjacency matrix-MP topology).** The analysis was run by randomly dividing the entire time span of evolution of the *Liolaemus alticolor- bibronii* group in three periods (18–5.5 Myr; 5.5–1.25 Myr; 1.25 Myr-present). Pie charts of each node depict the relative probabilities of ancestral area/ranges (showed in Legend). Letters above nodes represent ancestral ranges. Circles around pie charts represent events: blue circle: dispersal event; green circle: vicariance. See material and methods for area names shown in South America map. Time axis (in Myr) is annotated with major geological events.

Relationships within the *Liolaemus alticolor-bibronii* group were previously studied by *Quinteros (2013)* and *Aguilar et al. (2013)*. In his research, *Quinteros (2013)* includes 33 terminal taxa which belong to the *L. alticolor-bibronii* group, performing a more inclusive study. On the other hand, the molecular-based phylogeny by *Aguilar et al. (2013)* was focused on Peruvian terminal taxa (including eight additional taxa members of the *L. alticolor-bibronii* group). *Aguilar et al. (2013)* described the *L. alticolor-bibronii* group as paraphyletic (see below). *Morando et al. (2004)* and *Martínez et al. (2011)* performed phylogeographic analyses focusing on populations of *L. bibronii* and *L. gracilis,* and *L. bibronii,* respectively. The relationships found in the present study are similar to those detected by *Morando et al. (2004)* and *Martínez et al. (2011)*, despite the higher number of terminal taxa included.

The topology recovered by *Quinteros (2013)* showed low support and low resolution in the clades inside the *Liolaemus alticolor-bibronii* group. Nevertheless, we agree with

**Table 5  Mean node ages (Ma) and 95% highest posterior density interval (HPD) obtained for the *L. alticolor-bibronii* group and the main clades inside.** The ancestral areas inferred with a probability ≥ 0.5 with the unconstrained adjacency matrix are also shown. See Fig. 5 for location of groups.

| Group | Bayesian Inference | | Maximum Parsiomny | |
|---|---|---|---|---|
| | Age (95% HPD) | Ancestral Area/range | Age (95% HPD) | Ancestral Area/range |
| *Ancestor of the L. gravenhorsti and alticolor-bibronii groups* | 14.02 Myr (18.20–10.5) | BJ; 1 | 15.25 (20.08–9.50) | FJ; 0.65 |
| *L. alticolor-bibronii* | 12.84 Myr (20.01–5.64) | BL; 0.67 | 14.05 Myr (23.56–4.54) | F; 0.41 |
| *L. walkeri* | 11.54 Myr (20.26–2.8) | B; 1 | 12.78 Myr (19.65–5.87) | BF; 0.87 |
| *L. gracilis* | 10.49 Myr (17.06–3.85) | L; 0.5 | 12.39 Myr (17.42–7.25) | L; 0.5 |
| *L. robertmertensi* | 9.37 Myr (15.68–3.01) | HL; 1 | 4.2 Myr (6.84–1.67) | B; 0.36 |
| *L. bibronii* sensu stricto | 4.68 Myr (7.25–2.11) | L; 1 | 9.82 Myr (11.85–5.78) | L; 0.57 |

(*Quinteros, 2013*) who detected the *L. lemniscatus* and the *L. robertmertensi* groups nested inside the *L. alticolor-bibronii* group (Parsimony tree, Fig. 3). Previous studies found both groups outside of the *L. alticolor-bibronii* group (*Cei, 1986*; *Cei, 1993*; *Lobo, 2001*; *Lobo, 2005*; *Díaz Gómez & Lobo, 2006*). *Schulte et al. (2000)*, *Schulte (2013)*, *Pyron, Burbrink & Wiens (2013)*, while *Zheng & Wiens (2016)* found *L. robertmertensi* settled inside the *L. alticolor-bibronii* group, but recovered the *L. lemniscatus* group outside. Our BI tree is congruent with the latter topologies. The *L. alticolor-bibronii* group contains in two main clades (the *L. gracilis* and *L. walkeri* groups), which was not recovered previously. Reviewing the literature, the most similar topology is described by *Aguilar et al. (2013)*, which recovered a Peruvian clade (similar to our *L. walkeri* clade), a clade with species members of the *L. alticolor-bibronii* group and of the *L. monticola* and *L. pictus* groups.

The *Liolaemus walkeri* group recovered here shows the same composition as the Peruvian clade of *Aguilar et al. (2013)*, as well as species distributed in Bolivia and northern Argentina (Figs. 2 and 3). As we briefly mentioned above, the main differences between our parsimony and BI trees is the location of the *L. lemniscatus* group. Using parsimony, this group is recovered inside the *L. alticolor-bibronii* group and nested within the *L. walkeri* group, whereas in the BI tree it is located outside the *L. alticolor-bibronii* group as a sister of the *L. gravenhorsti* group. The topology recovered under BI is more congruent regarding species distribution, since the species of the *L. lemniscatus* group are usually found in central Argentina and Chile (as are the species members of the *L. gravenhorsti* group). The species of the *L. walkeri* group, on the other hand, are distributed in northwestern Argentina, Bolivia, and Peru.

The *Liolaemus gracilis* group has not been previously proposed as such. Again, the BI tree shows more biogeographical congruence than the parsimony tree. Nevertheless, the two main clades (*L. bibronii sensu stricto* and *L. robertmertensi* groups) which form the *L. gracilis* group are recovered in both analyses. In addition, the BI tree recovers *L. gracilis*,

*L. robertmertensi*, *L. sanjuanensis*, *L. saxatilis*, and *L. tandiliensis* as closely related (as in *Morando et al., 2004*; *Martínez et al., 2011*; *Aguilar et al., 2013*; *Pyron, Burbrink & Wiens, 2013*; *Zheng & Wiens, 2016*).

The greater congruence between the BI and previous studies, compared to the parsimony tree, can be explained by similarities of the dataset, given that they share the same gene sequences (*Morando et al., 2004*; *Martínez et al., 2011*; *Aguilar et al., 2013*).

The *Liolaemus lemniscatus* group was recovered inside the *Liolaemus alticolor-bibronii* group (nested in the *L. walkeri* group) under MP, but outside (sister of the *L. gravenhorsti* group) under BI. In previous molecular-based phylogenies, members of the *L. lemniscatus* group were found more related to species members of the *L. nigroviridis* and/or *L. monticola* group (*Panzera et al., 2017*; *Pyron, Burbrink & Wiens, 2013*; *Schulte et al., 2000*; *Schulte, 2013*). Moreover, *Pyron, Burbrink & Wiens (2013)*, *Schulte et al. (2000)*, and *Schulte (2013)* recover the *L. alticolor-bibronii* group as sister of the *L. gravenhorsti* group (*L. chiliensis* group of *Schulte, 2013*), with both more closely related to the *L. elongatus-kriegi* complexes than to the *L. lemniscatus* and *L. nigroviridis* groups. In our analyses under BI (excluding morphology data), the *L. lemniscatus* group is paraphyletic and is recovered outside the *L. alticolor-bibronii* group. Unfortunately, we cannot include species members of the *L. nigroviridis* and/or the *L. monticola groups* in the present study. This will be are search avenue to further investigate in future studies.

### Taxonomic implications

We included 15 populations of uncertain taxonomic status. These populations are currently assigned to known species members of the *L. alticolor-bibronii* group, but differ in some morphological character states. According to our study (BI and Parsimony trees), most of them can be considered candidate species. *Liolaemus* sp4 and *L.* sp5 are morphologically close to *L. araucaniensis*. In both trees, BI and Parsimony, *L.* sp4 and *L.* sp5 are not closely related to *L. araucaniensis* (which appears basal). This result, in addition to the varying character states, converts these terminal taxa into candidate species. *Liolaemus bibronii* was characterized as a species complex by *Morando et al. (2007)*, *Martínez et al. (2011)* and *Quinteros (2013)*. Over the last few years, three species previously confused with *L. bibronii* have been described: *L. cyaneinotatus* (*Martínez et al., 2011*), *L. abdalai* (*Quinteros, 2012*), and *L. yalguaraz* (*Abdala, Quinteros & Semham, 2015*). In this study, we include *L.* sp6, *L.* sp7, *L.* sp8, *L.* sp9, *L.* sp10 and *L.* sp15, all of which are assigned to *L. bibronii*. Some of these terminal taxa were included in the phylogeographic study of *Morando et al. (2007)*. Our results are consistent with those of *Morando et al. (2007)*, *Martínez et al. (2011)* and *Quinteros (2013)*, concluding that despite the newly described species, *L. bibronii* still corresponds to a complex of species with several candidate species. *Liolaemus robertmertensi* was described by *Hellmich (1964)*, but since its description no further taxonomic studies related to it. In this study, we included three populations attributed to *L. robertmertensi*: *L.* sp11, *L.* sp12, and *L.* sp13. *Liolaemus* sp12 has previously been included in molecular-based phylogenies (*Schulte et al., 2000*; *Espinoza, Wiens & Tracy, 2004*), whereas *L.* sp11 and *L.* s13 have only been included in a morphology-based phylogeny as *L. robertmertensi* (*Quinteros, 2013*). In both analyses (Bayesian and Parsimony) performed here, these terminal taxa are

not closely related to *L. robertmertensi*, while still being members of the *L. robertmertensi* group. *Liolaemus* sp14 corresponds to a population distributed near *L. tandiliensis*. This terminal taxon was discovered only recently and is currently under description.

## Inclusion of morphological and continuous characters in the analyses

We performed MP and BI analyses excluding morphological data (Figs. S4 and S5). Nevertheless, similar main groups are recovered. Under MP the *Liolaemus alticolor-bibronii* group is paraphyletic and includes members of the *L. gravenhorsti* group. BI topologies are similar to those detected under MP. The species members of the *L. gravenhorsti* group are nested inside the *L. alticolor-bibronii* group. Also, main clades of the *L. alticolor-bibronii* group are recovered, yet the relationships inside the *L. gracilis* group are not resolved. The *L. lemniscatus* group is recovered paraphyletic. Many authors mention homoplastic morphology (*Alvarez et al., 1999*; *Escobar García et al., 2009*; *Mott & Vieites, 2009*; *Mueller et al., 2004*; among others) but there is also evidence of highly homoplasious mtDNA data (*Engstrom, Shaffer & McCord, 2004*). *Jarvis et al. (2014)* performed a phylogenetic analysis in a very broad scale of major orders of birds 41.8 million base pairs (14536 loci) finding a topology with high values of bootstrap support. Gene trees (of every individual locus) in the same paper found that none of them was congruent to the species tree suggested by *Hahn & Nakhleh (2015)*. Here, the inclusion of morphological data increases clade resolution, and support the monophyly of some groups.

The study of continuous characters is limited to MP analysis, as it can only be performed by TNT software. Consequently, we tested which role continuous characters play in the recovered topology by excluding them. Without continuous characters the *Liolaemus alticolor-bibronii* group appears polyphiletyc. The topology recovers the main groups under MP and BI, but their composition include species members of the *L. gravenhorsti* group (see Fig. S6). The use of continuous characters analyzed as such was employed in many previous studies (*Abdala, 2007*; *Álvarez, Moyers Arévalo & Verzi, 2017*; *Barrionuevo, 2017*; *Bardin, Rouget & Cecca, 2017*; *Nájera-Cortazar, Álvarez Castañeda & De Luna, 2015*; *Quinteros, 2013*; among many others). In the present study, continuous characters employed in morphological analyses assisted to recover the monophyly of the *L. alticolor-bibronii* group and supported the monophyly of the *L. gravenhorsti* group. When comparing the results obtained under BI and MP (without continuous characters), we find that continuous characters as well as the optimality criterion influence the different topologies.

## The use of a priori component from DEC

Implementation of DEC models in Lagrange allows biogeographic analyses to infer biogeographic events under different probabilities, taking into account the distances among areas and the presence of geographic barriers. In this study, we applied two different adjacency matrices and three area-dispersal matrices, recovering different general patterns of diversification and species distribution in the *Liolaemus alticolor-bibronii* group, which allow inference of its evolutionary history.

*Chacón & Renner (2014)* explored the influence of the user-defined components applicable in DEC-Lagrange (adjacency matrix, area-dispersal matrix). They concluded

that results were most influenced by adjacency matrices, while the area–dispersal matrix and dispersal probabilities had negligible effects. In this study, we applied three different matrices with dispersal probabilities. Using the unconstrained adjacency matrix, the ancestral reconstructions were very congruent among each other; we found only one difference between the three matrices applied. In their study, *Chacón & Renner (2014)* found that an unconstrained adjacency matrix returned the best results (higher Likelihood score). Contrastingly, results by *Ree & Smith (2008)* and *Clark et al. (2008)* show a better data fit of the constrained matrix. In this study, we tested both, an unconstrained as well as a constrained adjacency matrix. The number of assigned ambiguous ancestral areas possible in each node was higher in the constrained than in unconstrained analyses (Table 4). Furthermore, the results obtained using the unconstrained matrix display higher Likelihood scores than the constrained matrix (Table 4). Based on these findings, we agree with *Chacón & Renner (2014)* concluding that the adjacency matrix influences the results in a DEC-Lagrange study, but at the same time, when applying a constrained matrix, the dispersal probabilities matrix had an effect on our results.

## Patterns of diversification and biogeographic implications

The use of two different phylogenetic topologies to estimate the time of divergences and reconstruction of ancestral distribution returned similar results (see Table 5). Despite the differences recovered in the composition of main groups, their divergence times were congruent. The exceptions were the *Liolaemus robertmertensi* and *L. bibronii sensu stricto* groups, even though an overlap exists. The ancestral area reconstruction also shows some differences between the topologies, but overall most results display great congruence. The main difference is in the ancestral area of the *L. alticolor-bibronii* group. The BI topology recovers the range BL (Puna and Patagonia Central), whereas the MP topology recovers the area F (Prepuna). The first one is congruent with previous studies in *Liolaemus* (*Cei, 1979*; *Díaz Gómez, 2011*; *Schulte et al., 2000*).

The fossil-calibrated dating analysis indicates that the initial divergence within *Eulaemus* occurred approximately 18.08 Myr ago during the Early Miocene, which roughly corresponds to previous studies (*Fontanella et al., 2012*). The origin and diversification of the *Liolaemus alticolor-bibronii* group could have started around the Early-Middle Miocene (13–14 Myr ago), which is consistent with results by (*Schulte, 2013*; *Medina et al., 2014*; *Zheng & Wiens, 2016*). During that period, many dramatic changes occurred, eventually leading to the Andes uplift. The Central Andes lifted quickly, geologically speaking, from the late Miocene to the Early Pliocene (*Whitmore & Prance, 1987*; *Gregory-Wodzicki, 2000*). The uplift separated the East from the West of the continent and affected the dispersion of many taxa and potentially induced the differentiation between populations (*Elias et al., 2009*). Our results support previous studies suggesting an influence of the Andean uplift on the diversification of South American taxa (*Schulte et al., 2000*; *Hoorn et al., 2010*; *Pincheira-Donoso et al., 2013*).

Regarding its evolution and distribution, it can be assumed that the *Liolaemus alticolor-bibronii* group colonized areas along the lower Andes. Our results indicate that the ancestral area of the group extended from the Patagonian Andes to the northern Andes (Argentina,

Bolivia, and Perú). This is in accordance with paleontological evidence: the oldest known fossil of *Liolaemus* corresponds to Patagonia in the Miocene era, at the Gaiman Formation in Chubut, Argentina (*Albino, 2008*).

Our findings are also consistent with those of *Díaz Gómez (2011)*, whose DIVA (Dispersion-Vicariance-*Ronquist, 1997*) analysis located the Liolaemidae common ancestor's distribution from Peru to Patagonia, along the Andes and arid regions of South America. *Cei (1979)* concluded that Patagonia was the center of origin for at least four *Liolaemus* groups, describing two main faunal regions of Patagonia: (1) septentrional region (older Patagonia) and (2) meridional region (Santa Cruz Province). Our results agree with *Cei (1979)*'s hypothesis as the first region described in his paper comprises areas of the Andes and Patagonia included in the distribution areas calculated in the present study. On the other hand, our results disagree with those published by *Schulte et al. (2000)* who found the areas of Andes, occidental lower lands, and eastern lower lands to be ancestral area of the *Liolaemus sensu stricto* subgenus which correspond to Sierras Pampeanas, Maulina, and Chile Central in our study.

According to *Bremer (1992)*, the ancestral area of a taxon is not necessarily to a single place, but may be equal or larger than the area presently inhabited by the taxon. Results of an ancestral area analysis will display a taxon's ancestral distribution range which is equal or smaller than the sum of distribution ranges of its descendants. Including a fossil group neither distributed in ancestral nor current areas of the studied taxa, may result in an ancestral area larger than the sum of the descendants' distributions. In our study, we used a fossil record attributed to *Liolaemus* (*Albino, 2008*) in an area already present in the analysis.

The common ancestor of the clade formed by the *Liolaemus alticolor-bibronii* group plus (*L. gravenhorsti* + *L. lemniscatus* groups) had its divergence around 14–15 Myr ago. This calibration coincides with the beginning of the Andes uplift (*Donato et al., 2003*) and our DEC analyses (Figs. 7–8), which suggest that vicariance event could have led to the current clade distribution in Chile (*L. gravenhorsti* and *L. lemniscatus* groups) Argentina, Bolivia, and Peru (*L. alticolor-bibronii* group). Because of their geography, orogeny, and vast biodiversity, the Andes Mountains were a subject of interest of many distribution and evolution studies (*Castroviejo-Fisher et al., 2014*; *Chaves & Smith, 2011*; *Goicoechea et al., 2012*; *Torres-Carvajal et al., 2016*). Based on a phylogenetic analysis of *Proctoporus/Riama* lizards, *Doan & Schargel (2003)* proposed the south-to-north speciation hypothesis (SNSH), which predicts a pattern of cladogenesis of Andean species following the rise of the Andes, with basal lineages occurring in southern areas and derived ones in northern areas. Our results gained with BI topology agree with the hypothesis of *Doan & Schargel (2003)*, since the ancestral area recovered for the *L. alticolor-bibronii* group includes Patagonia, and the derived groups show a northern distribution. On the other hand, our results obtained with MP topology reject the SNSH, since the ancestral range of the *L. alticolor-bibronii* group corresponds to Prepuna, and the derived groups are distributed across both, north and south of Prepuna. The SNSH was also rejected for *Proctoporus* lizards based on a phylogenetic analysis with increased character and taxon sampling (*Goicoechea*

*et al., 2012*), as well as for other organisms such as glass frogs (*Castroviejo-Fisher et al., 2014*).

The *Liolaemus gravenhorsti* group has its origins 7.72 Myr (95% HPD: 9.32–6.21) ago matching the continued Andes uplift. At that time (around 6.2 Myr ago), lakes and fluvial deposits were affected by a deformation event, possibly the final uplift stage reaching the present average altitude of 4,400 masl (*Garzione et al., 2008*), which may have triggered a dispersal or a vicariance event of *L. chiliensis* and *L. cyanogaster* towards Argentina.

The divergence time of the *Liolaemus alticolor-bibronii* group was estimated around 12–14 Myr ago, which corresponds to the formation of the Atacama Desert approximately 14.7 Myr ago, based on the lack of accumulation of cupric deposits in northern Chile (*Alpers & Brimhall, 1988*). Our results of BI topology suggest that following the desert's formation a vicariance event could have split the *L. alticolor-bibronii* group into two main clades distributed North (*L. walkeri* clade) and South (*L. gracilis* group) of the Atacama Desert. A similar scenario was recovered with MP topology, but instead of a vicariance event, dispersal events led to the *L. walkeri* group to the north, and the *L. gracilis* group to the south.

The diversification of the *Liolaemus gracilis* group approximately 10–12 Myr ago coincides with the uplift the Austral Andes (altitude >4,000 masl), which reached their maximum elevation around 8–12 Myr ago (Miocene-Pliocene) (*Hartley, 2003*). This geological event may have dispersed the species of the *L. gracilis* group giving origin to two clades: one distributed in central-northern Argentina (*L. robertmertensi* group), and a clade distributed in central-southern Argentina (*L. bibronii sensu stricto* clade). Similarly, *Cosacov et al. (2010)* report a phylogeographic break for *Calceolaria poliriza* in southern Mendoza possibly caused by a landscape discontinuity (*Ramos & Kay, 2006*) during the uplift of the Andes in the Late Miocene (11 Myr). Inside the *L. gracilis* group, the *L. bibronii sensu stricto* clade originates 5 Myr ago, while the group's more terminal clades diverge in the Pleistocene (2.5 Myr) and are distributed in Patagonia. *Sérsic et al. (2011)* studied phylogeographical patterns of plants and vertebrates (including some *Liolaemus* species) from Patagonia. Specifically, *Sérsic et al. (2011)* argued that the phylogeographic break of *L. bibronii* could be concordant with the Rio Limay basin which drains (or that in the past drained) the East Andean watershed crossing the Patagonian steppe to the Atlantic Ocean. Other studies (*Morando, Ávila & Sites Jr, 2003*; *Morando et al., 2007*; *Ávila, Morando & Sites Jr, 2006*) claimed that three species of *Liolaemus* (among them *L. bibronii*) shared a vicariant pattern produced by the coastline shifts along the Atlantic at the northern and southern rim of the Somuncurá Plateau.

The *Liolaemus robertmertensi* group (recovered under BI) is distributed in Sierras Pampeanas (Central and northwestern Argentina) and Sierras Subandinas (northwestern Argentina). A possible explanation for the present distribution of the group's Astereceae species is proposed by *Crisci et al. (2001)*: they found a historical pattern that related Tandilia to Ventania, Mahuidas, Sierras Pampeanas and Sierras Subandinas to the West, and Uruguay and southern Brazil to the East. *Crisci et al. (2001)* hypothesize that the endemism of these mountainous chains is a result of generally arid conditions during the Tertiary and/or Quaternary geologic periods in southern South America, which eventually

led to an isolation and differentiation of these populations in the more elevated areas. The ancestral area recovered under MP for the *L. robertmertensi* group is Puna, and its diversification time is approximately 4.2 Myr. This species group currently inhabits the Sierras Subandinas. These mountains showed a second deformation cycle at 4.5 Myr, with an out of sequence growth event (*Hernández & Echavarria, 2009*). This can explain the vicariance and dispersal events that the species of the group experienced.

The *Liolaemus walkeri* clade started its diversification 11–12 Myr ago in an area which today corresponds to the Andean Plateau. This plateau rose from 2,500 masl to 4,000 masl around 10 Myr ago, during the Miocene-Pliocene (*Gubbels, Isacks & Farrar, 1993*; *Gregory-Wodzicki, 2000*; *Hartley, 2003*) under an arid to semiarid climate conditions (*Hartley, 2003*). The *L. walkeri* clade is formed by two clades distributed in central and southern Peru, Bolivia and northern Argentina. This distribution could be due to a dispersal event from the Puna to Yungas (the northern distributed clade) while the other clade remained in the Puna. The clade split follows a similar pattern as the plant genus *Distichia* which separated due to the plateau's climate conditions into clades in northern Argentina and Chile (towards the South) and Ecuador and Colombia (towards the North) (*Ramirez Huaroto, 2012*).

The desertification process from the Miocene to Pliocene may have caused the range expansion of the ancestors of the species members of the *Liolaemus alticolor-bibroni* group. Subsequently, the arid/humid cycles which followed the glacial and interglacial period of the Pliocene and Pleistocene produced expansion and retraction of arid and damp habitats, acting as geographic barriers and causing fragmentation and speciation of the extant taxa.

In conclusion, we describe the possible ancestral area of the *Liolaemus alticolor-bibronii* group as a large area which includes Patagonia and the Puna highlands during the Lower Miocene and the Pleistocene. The inclusion of species members of the *L. monticola* and *L. nigroviridis* groups will probably modify the ancestral area-range of the *L. alticolor-bibronii* group.

## ACKNOWLEDGEMENTS

We are grateful to F Lobo, JM Díaz Gómez, C Abdala, M Quipildor, S Ruiz, T Hibbard, and F Arias for their assistance in the field, lab and/or discussion of ideas related to this study. We thank T Hibbard and M Schulze for improving the English style. We would like to thank S Kretzschmar, G Scrocchi and E Lavilla (FML); J Williams (MLP); J Faivovich, and S Nenda (MACN); JC Acosta, A Laspiur, and E Sanabria (UNSJ); L Vega (UNdMP), R Aguayo and F Valdivia (CBGR); J Aparicio, A Kirigin, and M Ocampo-Vallivian (CBF); H Voris and A Resetar (FMNH); L Ford and D Frost (AMNH); J Cadle and J Rosado (MCZ); J Wiens and E Censky (CMNH); T Reeder and R Etheridge (SDSU); K De Queiroz and R Heyer (USNM), for allowing access to their collections. We thank Willi Heniing Society for making TNT freely available. We thank two anonymous reviewers whose comments greatly improves the MS.

### Funding

This study was supported by grants from the Agencia Nacional de Investigaciones Cientificas (PICT-2015-1398) Consejo de Investigaciones de la Universidad Nacional de Salta (CIUNSa, 2013) and by a Graduate Fellowship from Consejo Nacional de Investigaciones Científicas y Técnicas (S Portelli). The funders had no role in study design, data collection and analysis, decision to publish, or preparation of the manuscript.

### Grant Disclosures

The following grant information was disclosed by the authors:
Agencia Nacional de Investigaciones Cientificas: PICT-2015-1398.
Consejo de Investigaciones de la Universidad Nacional de Salta (CIUNSa, 2013).
Consejo Nacional de Investigaciones Científicas y Técnicas.

### Competing Interests

Sebastián Quinteros is a researcher of IBIGEO-CONICET, Associate Editor of Cuadernos of Herpetología, and Secretary of the Asociación Herpetológica Argentina. Sabrina Portelli is a Doctoral scholar of CONICET.

### Author Contributions

- Sabrina N. Portelli and Andrés S. Quinteros conceived and designed the experiments, performed the experiments, analyzed the data, contributed reagents/materials/analysis tools, prepared figures and/or tables, authored or reviewed drafts of the paper, approved the final draft.

### Data Availability

    The raw data is included in the Supplemental Files.

### Supplemental Information

Supplemental information for this article can be found online at http://dx.doi.org/10.7717/peerj.4404#supplemental-information.

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
