# Peer review of "Phylogeny, time divergence, and historical biogeography of the South American Liolaemus alticolor-bibronii group (Iguania: Liolaemidae)"

_PeerJ, doi:10.7717/peerj.4404_

## Round 0.1 · original submission · Major Revisions

The reviewers both see merit in your paper and recommend that it may be suitable for publication.

However, there are a number of issues that are raised regarding the analysls. These are the choice of methods used for making the tree, the use of the fossil for,calibration and the combining of DNA and morphological evidence into a single tree. All these are valid points and should be addressed in any revision.

There are a number of other points made by reviewers, which would all need to be satisfactorily addressed.

Please also get your paper edited by a native English speaker before resubmitting.

Reviewer 1 ·

Basic reporting

The manuscript is well written in general but I think it would be a good idea to be corrected by a native English speaker. The background included in the text allows understanding how this work fits in the broader field of knowledge. The relevant prior literature is appropriate referenced. The general structure of the article is correct. The article is accompanied with supplementary material. However, the matrices (morphology and DNA) and obtained trees are not available in supplementary material. This should be corrected. The article include all the results required to answer the hypothesis (i.e. it is self-contained).

Experimental design

I have two major concerns about the analyses. One of them is related to the calibration point. The authors stated that:

“We used the fossil assigned to Eulaemus described by Albino 135 (2008), representing the earliest record of this subgenus to place a mean prior of 20 Myr on the tree height.”.

This seems not to be very good decision. If all that is known about the fossil is that it belongs to the subgenus, it might perfectly be place within any of the clades within it. In my opinion it is imperative to include the fossil in the phylogeentic study in order to use its age to define the mean prioir of the node that subtends its position (instead of directly assuming a basal position). Given that this is the only calibration point, taking care in this step is crucial to support the conclusions derived in this manuscript.



The second concern is related to the topological uncertainty. Although the trees obtained by maximum parsimony and Bayesian inference are similar, they differ in several nodes. However, the biogeographic and divergence time analyses were performed considering only the BI topology. This is not a good methodological practice. If both analyses were performed it is because the authors consider that the methods are methodologically sound. As a consequence, deriving conclusions from only one of the topologies in not correct. In my opinion the biogeographic and divergence time analyses should be repeated considering both topologies, showing the common patterns and differences between both results.

Validity of the findings

The validity of the findings is tied to what was expressed in the previous point. The authors should reanalyze the data and show that the methodological issues are not affecting the findings.

Additional comments

1-. Please include a paragraph in the main text describing how many new observations (i.e. scorings) were included in the morphological matrix.

2-. Line 114. If you used the free version of TNT you should cite the paper describing the program and acknowledge Willi Hennig society for making it freely available (see TNT documentation)

3. Line 115. Replace “k” by “concavity value”. Add implied weighting cite (Goloboff 1993).

4-. Line 229. Please include the details of this in M&M section not in discussion.

5-. Please rerun MP analysis without continuous characters in order to determine if the differences in the results obtained by MP an BI are related to the inclusion/exclusion of these characters.

6-. Figure 2. “The L. lemniscatus clade is recovered as a sister group of the L. walkeri clade under MP”
This statement is not compatible with the tree shown in Figure 4. Species of L. Lemniscatus are either nested within L. walker group or form a grade at the base of the tree.

The legend in Fig 4 is also incompatible with Figure 4, since not all the species of L. lemniscatus group are nested within L. walkeri group. (following the group definition of Figure 3):

7-. The probabilities calculated are only valid if the topology is correct. Given that you obtained a different topology with CC and Mp Please make it explicit

8-. L185. “There was high congruence between both phylogenetic analyses performed (Maximum Parsimony and Bayesian Inference), especially in the main groups (Figure 2)”. Please support this statement quantitatively: calculate SPR or number of shared nodes between them.

Reviewer 2 ·

Basic reporting

The article is a thorough evaluation of the systematics and biogeographic history of the Liolaemus alticolor-group, one the most widely distributed groups within Liolaemus. The paper is thorough, it is generally well written and flows well. However, it does have some major methodological issues I think need to be addressed before publication, so my recommendation is that this article might be suitable for publication in PeerJ after major revisions.
My main concern is the methodologies used, particularly to infer the phylogenetic relationships, combining morphological and molecular characters when these two have been found to be very incongruent in Liolaemus. More details are found below.

Experimental design

Introduction:

Line 53: This sounds odd: “…studies were performed with the L. alticolor-bibronii group as the aim of study”. Maybe replace to something like “L. alticolor-bibronii as the focal group”.

Line 68: Please give a reference for Liolaemus inhabiting 6,000 masl.

Paragraph from lines 71-86 should focus more on the results rather than the methodologies of those studies. What have they concluded, is there a consensus, what do they disagree on.

Line 65: Says the alticolor-bibronii group has 30 species but in line94 it says you inclided 31 species in the paper.

Materials and methods:

Paragraph in lines 94-97 is oddly written, please restructure the use of punctuation to make it more clear.

Line 100: web address of Genbank not necessary.

Line 103: was there any visual inspection of the alignments?

Line 123: Partitioning data might not be the best option with little molecular data, for this the best partitioning scheme must be tested with programs such as PartitionFinder.

Line 127: How did you conclude GTR+G was the most appropriate model for these data? Did you do any model test? I suggest you use JModelTest or PartitionFinder for this.

Did you assess convergence and mixing of the MCMC chains? It is very important to make sure all of the chians and also independent runs converge on the same area of probability space. You can assess this with Tracer or the RWTY package in R.

I think you should run a Bayesian Inference Analysis with the molecular data only, since it is often found in Liolaemus that morphological characters and molecular characters lead to very conflicting results. Morphological characters can be extremely linked and very prone to homoplasy, so phylogenetic inference on them should be taken with caution. This is not to say DNA is a perfect solution, but has fewer problems and is more informative.

Line 138: If you used the fossilized birth death then you should have included the fossil Liolaemus as a taxon in the set and not a calibration point. Otherwise you should stick to a model like calibrated Yule.

Line 142: I find unexpected that you reached proper convergence and mixing in BEAST after 5 million generations, this usually happens at list after 50 million and usually after 100 million, especially if you are doing a dated analysis with complicated models that have lots of parameters. I suggest rerunning this for 100.000.000 generations, or at the very least 50.000.000.

Line 143: Before combining chains you need to confirm that they have converged to similar values (looking at the trace plots in Tracer is the most common option).

Line 167: Did you compare the likelihood of other models, is there any reason to believe DEC is the most appropriate one?

If you are going to use Maximum Parsimony to infer your phylogeny, why not Maximum Likelihood as well?

Results:

Paragraph lines 187-193: I think it is very important that you run separate analyses with the mitochondrial loci and the morphological characters besides the combined, because they are both going to give you very different results. The lemniscatus group is found to be inferred as part of the nigromaculatus and not the chiliensis (where the alticolor-bibronii group sits) by molecular evidence, and that should be reflected in this paper, since it is going to give you substantially different biogeographic results.

Validity of the findings

Discussion:

In your initial paragraphs you are discussing the different and congruences between your results and previous studies. You should mention that the main difference is that your study agrees with others that use morphology but disagrees, mainly on the placing of L. lemniscatus, with the ones that are based on molecular evidence. This is why I stress that you should run independent phylogenetic analyses separating molecular from morphological evidence, and discuss the implications and drawbacks of both and of using combined data.

You need to compare the biogeographic models with different adjacency matrices in a likelihood framework, either by Likelihood ratio tests or by ranking them by AICc (corrected Akaike Information Criterion)

References:

Check for some errors, for example line 645: “Systematic Biologic”

---

## Round 0.2 · Minor Revisions

The paper has been substantially improved in line with the reviewer comments. However, reviewer 1 still has some issues with the calibration and with the discussion around congruence (or lack of it) between methods. These concerns need to be addressed satisfactorily before the paper can be accepted.

Reviewer 1 ·

Basic reporting

The authors have followed many reviewer’s suggestions that in my opinion have clearly improved the ms. However, I still have some concerns about the analyses. One of them is the calibrating point. I have expressed my concern about assigning a mean age of the basal node considering a fossil whose position in the tree is not clearly determined (only its placement as part of the subgenus is determined). If this fossil is in fact placed as sister of particular clade within the subgenus, all the ages estimated will be higher. The authors should at least be explicit about this point in the manuscript.
As suggested, the authors have repeated both the dating and the biogeographical analyses considering parsimony and MI topologies. However, the biogeographical discussion is still based only in the BI topology. Using the likelihood of the biogeographical analyses to choose the phylogenetic hypothesis is not a valid procedure. In my opinion, the discussion should explicitly include references to results that are incongruent in both analyses.
Finally, having a SPR distance of about 0.6 cannot be considered as supporting “congruence between topologies”. Indicate that there is congruence in the major groups but clear differences within those groups that produces that level of incongruence between parsimony and BI.

Experimental design

Indicated above and in the original review

Validity of the findings

Indicated above and in the original review

Additional comments

Indicated above

Reviewer 2 ·

Basic reporting

I am satisfied with the revisions done by the authors to mine and the other reviewer's comments. I think the paper has improved substantially and is more methodologically robust.

Experimental design

I think the design has improved and is sound now.

Validity of the findings

I think the conclusions are well supported by the data and results. I would though have some discussion on species not included in the analyses that belong to the group.

---

## Round 0.3 · Minor Revisions

This paper is very close. Just a few minor revsion are necessary. These are outlined in the attached PDF.

---

## Round 0.4 · accepted · Accept

Thanks for making these changes. The paper is now acceptable for publication.